# Traditional Human Immunodeficiency Virus treatment and family and social influence as barriers to accessing HIV care services in Belu, Indonesia

Nelsensius Klau Fauk[1,2]*, Lillian Mwanri[1], Karen Hawke[3], Paul Russell Ward[1]

1 Research Centre on Public Health, Equity and Human Flourishing (PHEHF), Torrens University Australia, Adelaide, South Australia, Australia, 2 Institute of Resource Governance and Social Change, Kupang, East Nusa Tenggara, Indonesia, 3 Infectious Disease - Aboriginal Health, South Australian Health and Medical Research Institute, Adelaide, South Australia, Australia

* nelsen_klau@yahoo.com

**Data Availability Statement:** All relevant data are within the manuscript.

**Funding:** The author(s) received no specific funding for this work.

## Abstract

Access to HIV care services, including antiretroviral therapy (ART), is essential for improving health outcomes of people living with HIV (PLHIV) and reducing HIV transmission and AIDS-related deaths. As a part of a qualitative study in Belu, this paper describes the use of traditional medicines for HIV treatment and family and social influence as barriers to access to HIV care services among PLHIV. One-on-one in-depth interviews were employed to collect data from 46 PLHIV (26 women and 20 men) and 10 healthcare professionals. They were recruited using the snowball sampling technique. The study information sheets were initially posted on information boards in healthcare facilities. Potential participants who contacted to confirm their participation were recruited for an interview and then asked for help to distribute the information sheets to their eligible colleagues who might be willing to participate. Data analysis was performed using NVivo 12 software and guided by a qualitative data analysis framework. The findings showed that the use of traditional medicines, a well-known cultural practice in Belu, was a barrier to access to HIV care services among PLHIV. The influence of family in determining the use of traditional medicines for HIV treatment, supported by the lack of knowledge of ART, effectiveness of traditional medicines in treating other health issues, and social influence of relatives, neighbours, and friends, were also significant barriers to PLHIV's access to HIV care services. The findings indicate the need for dissemination of HIV care-related information for PLHIV, family, and community members to increase their knowledge of the service, ART and its function, and to support and improve access to HIV care services especially ART by PLHIV.

## Introduction

Access to HIV care services, including antiretroviral therapy (ART), is essential for the improvement of health outcomes of people living with HIV (PLHIV). During the period from 2010 to 2019, ART was reported effective in reducing HIV infections by 23% and AIDS-related

**Competing interests:** The authors have declared that no competing interests exist.

**Abbreviations:** AIDS, Acquired immune deficiency syndrome; ART, Antiretroviral therapy; HCPs, Healthcare professionals; HIV, Human Immunodeficiency Virus; PLHIV, People living with HIV; THs, Traditional healers; TMs, Traditional medicines.

deaths by 39% globally [1]. However, during the same period, Indonesia experienced a significant increase in the burden of HIV, including an increase of up to 132% infections and 60% AIDS-related deaths, a reflection of the inequitable distribution or limited coverage of ART, late diagnosis, and poor access and adherence to ART [1, 2]. The 2021 national AIDS report shows that of the total number of 427,201 PLHIV in the country, only 63% ever started ART [2]. Of the ones who have started ART, 26.9% have failed to follow up or have stopped the therapy, 18.3% have died, and 53.7% are currently on ART [2].

Globally, previous studies have reported a range of barriers to access to HIV care services among PLHIV. These include limited availability of HIV care services, the shortage of qualified healthcare professionals (HCPs) to deliver the services to PLHIV, long-distance travel to healthcare facilities or HIV clinics and the lack of public transportation [3–7] A limited approachability of HIV care services reflected in poor dissemination of information about the services and the poor health literacy of PLHIV about both HIV and HIV care services, are also barriers for them to perceive their needs for care and access the services [8–10]. Poor health literacy is also reported to influence them in making critical decisions about their health, including seeking and accessing available and appropriate healthcare services [5, 11–13]. The unaffordability of HIV care service-related costs and the inability of PLHIV to afford them due to poor economic conditions are also barriers to their access to the services [5, 9, 14–17]. Concerns about the confidentiality of HIV status, fear of losing a job if HIV status is known to others, perceived healthy status, lack of time, and psychological burden of undergoing HIV care, are reported as demotivating factors for their access to HIV care services [10, 15, 18–21]. Similarly, perceived, anticipated and external stigma from families, community members, and healthcare professionals are also barriers to their access to the services [22–27].

Despite a range of barriers as reported in the aforementioned studies, the literature suggests that evidence on the influence of cultural practices (e.g., the use of traditional medicines (TMs)), family and social factors on access to HIV care services among PLHIV is still limited. TMs in the context of Belu refer to the roots, leaves, and barks of plants that are prepared and mixed by traditional healers (THs) or family members and used for the treatment of HIV in PLHIV [28]. THs in this context refer to people who do not attend formal medical training but are considered by family or community members as competent to provide HIV care using roots, leaves, and barks of plants based on traditional practices [29]. Previous studies in Africa have reported the use of TMs and their influence on ART adherence among PLHIV [30–34]. However, none of these studies specifically focused on exploring the use of TMs as a barrier to access to HIV care services among PLHIV. For example, studies in Ethiopia and Uganda reported on the common concomitant use of TMs and ART as a barrier to ART adherence [31–33]. Similarly, a study in Tanzania, Uganda, and Zambia reported on consultation with a TH as a factor associated with poor ART adherence [30]. Only a recent study in Ethiopia by Gesesew and colleagues, which explored the access of women living with HIV (WLHIV) to HIV care services, reported on the restrictions of THs towards the use of ART as barriers for the women to accessing HIV care services [35]. This paper aimed to fill this gap in knowledge by exploring the cultural practice of the use of TMs for HIV treatments in Belu, and family and social influence on HIV treatment of PLHIV. It also explored how these barriers to HIV care access had impacted health outcomes and the life of PLHIV. Our findings will be useful information for government and non-governmental institutions, policy makers, and program planners to address the issue through policies and interventions for better health outcomes of PLHIV in Indonesia and globally.

## Methods

The report of the methods section was guided by consolidated criteria for reporting qualitative studies (COREQ) checklist [36]. The checklist contains 32 items that need to be covered to support the explicit and comprehensive reporting of qualitative studies [36].

### Study setting

Belu is a district in East Nusa Tenggara province, located in the eastern part of Indonesia. It shares the border with East Timor with a total population of 204,541 people [37]. The district comprises 12 sub-districts and had three hospitals, 17 public health centres located in each sub-district, 21 sub-public health centres, 23 village health posts, 48 village maternity posts, five private clinics, and one HIV clinic where ART is provided. It has a total number of 1,200 PLHIV, but just over half of them (637 people) had accessed HIV care service or ART. Of the ones who had accessed ART, only half regularly accessed and adhered to ART at the time this study was conducted [38]. ART was the only HIV care service available for PLHIV in Belu [38]. The use of TMs for the treatment of any health issues, including HIV, is also a common practice within families and communities in this area, which might have been a barrier to the access of PLHIV to HIV care services. To the best of our knowledge, there has not been any study exploring the influence of TMs and the social influence of families and friends on access to HIV care services among PLHIV in the context of Belu and Indonesia. Belu was selected as the study setting due to the feasibility, familiarity, and potential of undertaking the current study successfully.

### Recruitment and data collection

The participants were PLHIV and HCPs recruited using the snowball sampling technique. PLHIV and HCPs were respectively recruited from an HIV clinic and public health centres providing HIV care services in Belu. Following the initial talk and agreement with the heads of these healthcare facilities, the study information sheets for PLHIV and HCPs were initially distributed through the receptionists in these healthcare facilities. The information sheets which contained a brief explanation about the study and contact details of the field researcher (NKF) were posted by the receptionists on their information boards. Potential participants who called to confirm their participation were recruited and asked to suggest a preferred time and place for an interview. The initially interviewed participants were also asked for help to distribute the study information sheets to their eligible friends and colleagues who might be willing to participate. This process was iterative, leading to a total of 46 PLHIV (26 women and 20 men) and 10 HCPs participating in this study. Two people withdrew their participation after a few minutes of the start of the interview due to personal reasons and their information was excluded from this report.

One-on-one in-depth interviews were used to collect data from the participants. Interviews with PLHIV and HCPs were respectively conducted in a private room at the HIV clinic and at healthcare facilities where the HCPs worked, which was mutually agreed upon by the field researcher (NKF, male) and each participant. Only the researcher and each participant were present in the interview room. Interviews were conducted in Bahasa for a duration of 35 to 87 minutes and audio recorded. Notes were also taken during the interviews. Pertaining to the topic of traditional treatment, the interviews focused on several key areas, such as PLHIV's experiences with access to HIV care services; whether or not they regularly access the services; factors that influenced their access to the services; whether or not they used other treatment (e.g., TMs) in addition to ART; the influence of their family members, relatives, friends and neighbours in their HIV treatment; and HCPs' knowledge about barriers to access to HIV care

service by PLHIV. Recruitment of the participants and interviews ceased when the researchers felt that no new added information or data in the responses of the last participants, an indication of data saturation. Interviews were conducted in Bahasa Indonesia, the primary language of the participants and the field researcher who is from Belu and who can also speak English fluently. The field researcher is a PhD student in public health and had attended training on qualitative research through formal education. The participants (PLHIV) were not offered an opportunity to review their transcript to prevent the possibility of the transcript being received by their family members once it was sent to them in hard copies. This could lead to a breach of the confidentiality of their HIV status, in case they had not disclosed it to their family members. No repeated interview was conducted and no established relationship between the researcher and any participants prior to the interviews.

## Data analysis

The audio recordings were transcribed verbatim and manually using a laptop by the first author (NKF). Data analysis was performed using NVivo 12 software and guided by the five steps introduced in the qualitative data analysis framework by Ritchie and Spencer [39]. These steps are (i) familiarisation with the data or transcripts through reading the transcripts line by line and repeatedly, providing comments and labels, and highlighting ideas related to barriers to access to HIV care services; (ii) identification of a thematic framework by writing down key issues and concepts; (iii) indexing the data by creating open coding to each transcript, where data extracts of each transcript were given a code or node. This was followed by close coding to identify similar or redundant codes and group them together to reduce the length of the coding list. Codes that formed the same themes or sub-themes were grouped together; (iv) charting data by arranging a thematic framework in a summary of chart; and (v) mapping and interpretation of the data [39, 40]. Data analysis was primarily performed by the field researcher, although the team-based analysis was carried out at regular supervision meetings whereby comments and suggestions were provided and discussed, and then team decisions were made about the validity of the final themes and interpretation.

## Ethical consideration

The study was approved by the Social and Behavioural Research Ethics Committee, Flinders University, Australia (No. 8286), and Health Research Ethics Committee, Duta Wacana Christian University, Indonesia (No. 1005/C.16/FK/2019). In addition, a permission letter from the local government of Belu district was obtained prior to the data collection (No. RSU/890/Diklat/423/VIII/2019). All the study participants were informed about these ethics approvals and permission from the government prior to commencing the interviews. They were informed about the purpose of the study through the study information packs and by the research prior to the interview. They were also advised that their participation was voluntary and that they had the right to withdraw their participation for any reasons before or during the interview without any consequences. They were informed that the interviews will be digitally audio recorded and assured that the information they provided during the interview will be treated confidentially and anonymously by assigning each participant a specific letter and number. This helps to prevent the possibility of linking back the data to any individual in the future. Before commencing the interviews, each participant was provided with informed consent to sign and return to the research.

## Results

### Demographic profile of the participants

The participants' (PLHIV) age ranged from 18 to 60 years old, and half of them were married, while the others were unmarried. The majority of them (33 people) have been diagnosed with HIV for 1 to 5 years at the time this was conducted, while the rest have been diagnosed with HIV for 6 to 10 years (11 people) and 11 to 15 years (2 people). Several of them have also been diagnosed with other infections, such as herpes, candidiasis, gonorrhoea, and tuberculosis (see Table 1). Most participants graduated from either elementary school or high school (38 people), and only a few graduated from university (8 people). The majority of them had different kinds of jobs, a few reported being housewives (11 women) and some were unemployed (3 women). The age of the healthcare professionals ranged from 30 to 49 (8 people) and the majority of them were married (7 people). All of them graduated from university, eight people were nurses and two were medical doctors.

### Traditional treatment of HIV

**Traditional treatment as a cultural practice.**   The use of traditional medicines to treat any kind of diseases, including HIV/AIDS was common among community members in Belu. These medicines were made of a combination of roots, leaves, and the bark of plants, as one of the participants said "*There are various common traditional medicines which are made of the roots, leaves, and barks of plants, and those plants can be found around us*" (MP9, married). PLHIV acknowledged that the use of traditional medicines provided had been a well-known cultural practice within communities, passed down from one generation to another:

> "*It has been our culture since a long time ago, from our ancestors that if people get sick then they would firstly seek for traditional medicines*"
>
> *(FP15, widowed).*

> "*It (the use of TMs) is a common practice here (in Belu). People use different types of traditional medicines to treat their health problems. My parents, grandparents and so on have used traditional medicines for years even before I was born. . ..*"
>
> *(MP7, married).*

The practice of traditional medicine use for the treatment of health issues was also recognised by healthcare professionals interviewed in this study. They acknowledged the existence of such practice within communities in the study setting and even within their own families. All healthcare professionals described that the majority of PLHIV in the district who did not access HIV care services or medical treatment were still using traditional medicines for their HIV treatment:

> "*The use of traditional medicines is common and has been passed down from one generation to another until now. Those medicines exist within communities and families. Even my family members and relatives still use traditional medicines (for other health issues). So, I am not surprised that there are also traditional medicines for HIV treatment and many PLHIV are using those medicines*"
>
> *(HCP10, nurse and counsellor).*

**Table 1. Sociodemographic profile of the participants.**

| Characteristics | Women and men living with HIV | | Healthcare professionals |
| --- | --- | --- | --- |
| | Women (N = 26) | Men (N = 20) | (N = 10) |
| **Age** | | | |
| 18–19 | 2 | | |
| 20–29 | 4 | 7 | |
| 30–39 | 12 | 5 | 4 |
| 40–49 | 6 | 5 | 4 |
| 50–59 | 1 | 2 | 2 |
| 60–69 | 1 | 1 | |
| **Marital status** | | | |
| Single | 3 | 7 | 3 |
| Divorced | 1 | | |
| Widowed/r | 12 | 1 | |
| (Re)Married | 10 | 12 | 7 |
| **HIV diagnosis** | | | |
| 1–5 years ago | 18 | 15 | |
| 6–10 years ago | 7 | 4 | |
| 11–15 years ago | 1 | 1 | |
| **Other infections** | | | |
| Herpes | | 1 | |
| Candidiasis | 3 | | |
| Gonorrhoea | | 1 | |
| TB | 5 | 9 | |
| **Education** | | | |
| University graduate/Diploma | 6 | 2 | 10 |
| Senior High school graduate | 5 | 8 | |
| Junior High school graduate | 6 | 4 | |
| Elementary school graduate | 8 | 6 | |
| Elementary school dropout | 1 | | |
| **Occupation of PLHIV** | | | |
| Housewife | 11 | | |
| Entrepreneur | 6 | 1 | |
| Teacher/University student | | 3 | |
| Farmer | | 3 | |
| Nurse/health worker | 2 | | |
| Private employee | 2 | 3 | |
| Civil servant or retired | 2 | 2 | |
| Taxi/truck/ Motorbike taxi driver | | 8 | |
| Unemployed | 3 | | |
| **Occupation of HCPs** | | | |
| Medical doctor | | | 2 |
| Nurse | | | 8 |

*"The use of traditional medicines for HIV treatment is very common among PLHIV in Belu. I am sure that most PLHIV who have known their HIV status but do not start antiretroviral therapy are taking traditional medicines"*

*(HCP1, medical doctor).*

**The impact of traditional treatment use on HIV care access and the health of PLHIV.**
The availability and common use of traditional medicines for the treatment of health issues
within communities in this area seemed to have a significant influence on the participants' and
other community members' health-seeking behaviours. The narratives of the participants
revealed that traditional medicines had often been used in the first place for the treatment of
health issues by many community members in the district. The following quote reflects the
availability of traditional medicines and how it influenced health-seeking behaviours of com-
munity members, including himself:

> *"The use of traditional medicines is a common practice here. . . .. People use traditional medi-
> cines to treat HIV too. Once I was tested positive, the first treatment my parents thought of
> and prepared for me to use was traditional medicines. They searched for those roots, leaves,
> and barks of plants, and prepared for me. It is because traditional medicines are easy to make
> and are commonly used in our community. My parents have also been taking those kinds of
> traditional medicines. . .."*

> *(MP4, Belu).*

It was apparent that the cultural practice of traditional medicine use influenced or delayed
the acceptability of and access to HIV care services among both women and men living with
HIV in Belu. It influenced the initiation of and retention in HIV treatment. Nearly half of the
women and men interviewed described that they did not access the HIV care service straight-
away following their HIV diagnosis or stopped the biomedical therapy due to undergoing tra-
ditional treatment using traditional medicines:

> *"After the HIV diagnosis, the doctor told us (the woman and her husband) that the medicines
> for HIV treatment are available at this hospital (HIV clinic) but we did not access the medi-
> cines directly. We did the treatment using traditional medicines provided by a traditional
> healer in XX (name of a place)"*

> *(FP8, widowed).*

> *"After the diagnosis, I accessed the therapy (ART) and then once I finished the medicines (first
> month), I switched to traditional medicines. I came back here (to restart ART) because my
> physical condition was getting weaker"*

> *(MP9, married).*

Similar stories were also echoed by all the healthcare professionals interviewed in this
study. They described how the use of traditional medicines for HIV treatment influenced
access to HIV care services by PLHIV in the district:

> *"There are patients who do not use traditional medicines straightaway after the HIV diagnosis
> even though we (HCPs) have talked to them and encouraged them to start ART. Some
> patients have started ART for a few months and then quitted and switched to traditional
> medicines"*

> *(HCP4, nurse).*

The use of traditional medicines for HIV treatment caused negative impacts on the health
of PLHIV. Both women and men living with HIV, who previously used traditional medicines,

described that the use of traditional medicines for HIV treatment worsened their health. Similarly, the healthcare professionals commented the use of traditional medicines for the treatment of HIV or switching from antiretroviral medicines to traditional medicines often deteriorated the health condition of PLHIV:

> *"My husband and I took a traditional medicine, but my husband's condition got worse, so we went back to the hospital (to do continue treatment with ART, her husband died from AIDS)"*

> *(FP4, widowed).*

> *"There were also patients who have started the therapy (ART) for a few months and then quitted and switched to traditional medicines. Some of these patients came back to this clinic to restart ART once their physical and health conditions gradually declined. I have seen many HIV-patients whose (physical) condition got worse and HIV status progressed to AIDS level due to using traditional medicines instead of ART"*

> *(HCP3, nurse and counsellor).*

Furthermore, the use of traditional medicines for HIV treatment seemed to also lead to worse consequences such as the death of PLHIV. Some women and men living with HIV described the situation where their spouses who were also HIV-positive passed away during the treatment using traditional medicines, which in turn became the underlying reason for their return to access HIV care services at HIV clinics. Similarly, healthcare professionals acknowledged such fatal consequence of the use of traditional medicines on PLHIV in the district:

> *"After the HIV diagnosis, the doctor told us (the woman and her husband) that the medicines for HIV treatment are available at this hospital (HIV clinic) but we did not access the medicines directly. We did the treatment using traditional medicines provided by a traditional healer in XX (name of a place). . . .. As I said before, traditional medicines did not help, and our health condition declined. That was the reason why we decided to come here (HIV clinic) to access these medicines (ART) but my husband could not make it, he passed away"*

> *(FP8, widowed, Belu).*

> *"There have been many patients with HIV who died from AIDS because they did not want to comply with what we (HCPs) told them to do. We keep encouraging each of them until now to undergo ART and leave traditional medicines but there are still many who are holding on to their traditional medicines. You can see from the data, of more than one thousand people diagnosed with HIV, there are only around 300 who access and adhere to ART. Nearly every month we get the report from nurse or counsellor from public health centres at sub-district levels that patient A or B (PLHIV) has passed away"*

> *(HCP1, medical doctor).*

**Economic consequences of traditional treatment of HIV.** The use of traditional medicines was reported to have financial consequences for PLHIV in Belu as reported by both PLHIV and HCPs in this study. Some participants, both women and men living with HIV, acknowledged that they had to pay a certain amount of money and give a sacrificial animal to traditional healers who provided the traditional medicines. The amount of money and

the animal seemed to differ from one traditional healer to another and had to be provided prior to the commencement of the treatment. Meanwhile, some other participants revealed that they had the traditional treatment for free as the traditional healers were in their families:

*"The use of traditional medicines for the treatment of any kinds of diseases is very common here, but it is costly. Patients have to pay some amount of money and take with them a chicken or pig or goat as a sacrificial animal. I heard that to get traditional medicines from Marry (pseudonym of the traditional healer), a patient has to pay five million rupiahs and give a chicken or pig to her"*

*(MP16, married).*

*"I used traditional medicines from my relatives, they (husband and wife) knew that I am infected with HIV, and offered the medicines to me. . . .. I did not pay them, they are my family, they just wanted to help me and did not ask for money"*

*(FP7, remarried).*

The desire to recover faster and be completely cured of HIV, and distrust of HIV test results, were reported as barriers to the acceptance of medical treatment using antiretroviral medicines among PLHIV. These were acknowledged by the participants from both groups as supporting factors for the decision of PLHIV and their families not to access HIV care services or use medical treatment or switch from ART to traditional medicines:

*"To get the traditional medicines and undergo the treatment with the healer, we (the man and his wife who is also HIV-positive) had to pay a lot of money to him (the healer) and give him a goat. It was very expensive, but we had to do it because we wanted to get better faster. We spent about fifteen million rupiahs on this treatment. This amount had to be paid at once before the start of the treatment, only once. My family: parents and siblings helped me with the payment"*

*(MP11, married).*

*"One of the reasons they or their families decide to use traditional medicines or switch from antiretroviral therapy to traditional treatment of HIV is because of the desire to get better faster or full recovery. Some patients told me that they wanted to get cured and strong faster, that is why they switched to traditional medicines, even though we have told them that they will not be completely cured of HIV and therefore they have to take antiretroviral medicines every day for the rest of their lives"*

*(HCP9, nurse and counsellor).*

## The influence of family on the use of traditional medicines for HIV treatment

**Family decisions for traditional treatment of HIV.** The participants, both PLHIV and HCPs, reported that family members had a crucial role in determining the treatment for PLHIV. Several female and male participants (PLHIV) commented that they underwent HIV treatment using traditional medicines due to being asked by their family members, such as parents or in-laws. Such an influence of family members was also reported by HCPs:

*"After he (her late husband) was diagnosed with HIV, at first the doctor gave him cotrimoxazole for two weeks, but they (her husband's family) told him to take traditional medicines. . . .. They asked him and me to use the traditional medicines for bathing as well. My husband and I took the traditional medicines, but after a while, my husband's condition got worse, so we went back to the hospital (to start ART but her husband died)"*

*(FP4, widowed).*

*"I see that family members have a dominant role in determining treatment for the sick ones (HIV-positive). Often people are diagnosed with HIV while they are in a severe condition and being admitted to hospital. So, once they left the hospital, many of them switched to traditional medicines because their family members asked them to do so and then stopped taking antiretroviral medicines. Family members are the ones who look for traditional medicines for them (PLHIV)"*

*(HCP10, nurse, and counsellor).*

Both PLHIV and HCPs described that family members' decisions to use traditional medicines for HIV treatment of their sick family members were influenced by their experience with the effectiveness of traditional medicines in treating other health issues. Several participants (PLHIV) described that their family members were regular, long-term users of traditional medicines, and therefore asked them to use traditional medicines for HIV treatment. This also seemed to be supported by the lack of knowledge among their family members about HIV care services provided in the HIV clinic:

*"My parents and grandparents are very familiar with traditional medicines and these are the number one medicines for them. Every time they feel sick or unwell, they use traditional medicines to treat their body. So, they recommended the use of traditional medicines to treat HIV as well. They do not know anything about medical treatment like these (showing her antiretroviral medicines she just collected)"*

*(FP5, widowed).*

*"In general, (HIV) patients come from families with a low level of education. Their family members do not know about ART and do not understand the function of the therapy to suppress viral load. Besides, I believe their family members have seen and experienced healing from certain ailments or health issues due to the use of traditional medicines. I think, these are also the reasons why many families rely on traditional medicines for the treatment of their HIV-positive family member"*

*(HCP3, nurse and counsellor).*

Besides, the stories of some HCPs and PLHIV interviewed in this study showed that being physically weak and taken care of by their family members, and a lack of knowledge about HIV care services, especially ART, appeared to be personal-related supporting factors that made them accept and follow the recommendations of their family members for the use of traditional medicines:

*"Once we (the woman and her husband) were tested positive with HIV, our physical conditions were weak already, our families took care of us. My parents-in-law asked us to use traditional medicines. So, we just used them, my mother-in-law sent them to us every month. We*

*just listened to what they said and used the traditional medicines because we wanted this HIV to go away"*

(FP23, married).

*"It is true that family members play a very important role in the treatment of HIV patients. If their family members provide traditional medicines, then they (PLHIV) will definitely take the medicines. There is no way for them to refuse because they are taken care of by family members, and they are sick"*

(HCP7, nurse and counsellor).

**Social influence on the use of traditional treatment of HIV.** Extended family members and neighbours were also reported to have an influence on the decision of the participants' family members regarding the use of traditional medicines to treat HIV infection. Providing information about traditional medicines for HIV treatment, and encouraging the participants' family members about the effectiveness of traditional medicines to treat HIV infection, were in some instances reflecting the influences of others on the participants' family members regarding the use of traditional medicines:

*"My family members encouraged us (the woman and her late husband who died from AIDS) to use traditional medicines because they were encouraged by our extended family members and neighbours that traditional medicines can cure HIV*

(FP8, widowed).

*"We (the man and his wife) know about a traditional medicine for HIV from a friend of mine. He came to my house and talked to my wife and me about it. My wife was convinced by his story that traditional medicine is very good and has cured people of this disease (HIV infection). Then my wife encouraged me to switch (from ART) to that traditional medicine"*

(MP9, married).

## Discussion

Increased HIV transmissions and AIDS-related deaths in many countries and regions globally have been associated with poor access to HIV care services and poor adherence to ART [1]. This paper describes the influence of the use of traditional medicines for HIV treatment and the role of families, friends, and neighbours in determining HIV treatment for PLHIV, which have not been addressed in previous literature on barriers to access to HIV care services [41–44].

Previous studies have reported the existence of traditional medicines within communities in some African countries and their influence on ART adherence for PLHIV [30–34]. These were also apparent in our data from both women and men living with HIV and HCPs. What our data add is the concept of the use of traditional medicines for the treatment of HIV and other health issues as a cultural practice that has been passed down from one generation to another within communities in Belu. This concept also seemed to indicate how influential this practice is on the health-seeking behaviours and the preference for treatment by family members of PLHIV and other community members in the study setting. Our study adds further evidence suggesting that the use of traditional medicines for HIV treatment influenced or

delayed acceptability and access to biomedical HIV care services among both women and men living with HIV. Such an influence seemed to be strongly supported by the availability and approachability of traditional medicines within communities in the area. These support the constructs of access to healthcare service framework [5, 9], suggesting the availability of a healthcare service and approachability or how well-known the information about the service is among people in health needs as some of the dimensions determining the accessibility of the service. The current study also suggests that the use of traditional medicines for HIV treatment had a negative financial impact on some participants as the service costed large amounts of money and required sacrificing animals.

Previous studies have found that support from family members played an important role in the successful access to HIV care services by PLHIV and their retention in medical treatment or ART [45–47]. However, our findings suggest that the role of family members in determining the use of traditional medicines for HIV treatment was a significant barrier to access to HIV care services and the initiation of ART by PLHIV in this study. Factors such as the regular use of traditional medicines, the effectiveness of traditional medicines in treating other health issues and a lack of understanding about ART and its function were the underlying reasons for family decisions supporting the use of traditional medicines over ART. These are similar to previous findings which report insufficient or incorrect knowledge about ART as a barrier to access and adherence to ART among PLHIV [43], and positive relationships between a lack of ART knowledge and less family support for ART access and adherence [48]. In short, consistent with previous findings reported elsewhere [41–44], the current study suggests a lack of family support for the participants (PLHIV) prior to or in the early stages of their HIV treatment as a barrier to their access to HIV care services and treatment adherence.

The social influence of relatives, neighbours, and friends on family decisions regarding the use of traditional medicines for HIV treatment was another access barrier for PLHIV. These support the findings of a recent study suggesting that social influences of sexual partners, peers, and healthcare professionals are a hindrance to PLHIV seeking HIV treatment and care services [49]. Thus, our findings indicate that the role of good social support from people surrounding PLHIV is crucial for prompt and sustained access to HIV care services and treatment adherence, as has been reported elsewhere [46, 50–52]. In addition, evidence from the current study also demonstrates that the poor physical and health conditions of PLHIV, and their dependency on family support for their daily needs and healthcare influenced their ability to make decisions about their health treatment. This often led to them deferring to their family about treatment decisions, including the use of traditional medicines which negatively affected their willingness to access and adhere to ART.

## Study limitations and strengths

PLHIV who participated in this study were recruited from an HIV clinic in Belu, whilst some of them may have stopped ART at some point, they are still engaged in care. We did not include PLHIV who are disengaged from care, who may have had different stories to tell about the impact of family or traditional medicines on their use of ART. However, to our knowledge, this is the first qualitative inquiry exploring the influence of traditional medicine use and the role of families on the access to HIV care services among PLHIV in the context of Indonesia. Our findings have important implications for the health sector in Belu and other similar settings in Indonesia. As new HIV infections and AIDS-related deaths significantly increase every year, there is a need for responses at the policy level and the development of interventions that address barriers to access to HIV care services experienced by PLHIV. Such efforts may lead to an expansion of the coverage of HIV care services especially ART and increased

access to the services. The current findings also have an implication for policy and practice in Belu district and other similar settings in Indonesia and globally to address and develop HIV information and education programs for family and community members to improve their health literacy about HIV, ART and its function. As family members have an important role in the HIV treatment for PLHIV, HIV education programs may result in better understanding and access to HIV care services and retention in HIV treatment among PLHIV.

## Conclusions

The paper reports the use of traditional medicines for HIV treatments and the role of family members in determining treatment for their HIV-positive family members as barriers to access to HIV care services among PLHIV in Belu. It shows that the use of traditional medicines in treating any kind of health issues, including HIV is a well-known cultural practice within communities in Belu, which influenced participants' (PLHIV) access to HIV care services and ART adherence. It also reports a strong family role in determining the use of traditional treatment for an HIV-positive family member, which was supported by a lack of knowledge about ART, positive experience of the effectiveness of traditional medicines in treating other health issues and the social influence of other. Future large-scale studies are recommended to obtain a comprehensive understanding of the influence of cultural, family, and social factors on access to HIV care services among PLHIV in Belu and other settings in Indonesia and globally.

## Supporting information

**S1 Fig. Questionnaire-typeset39MIO.**
(DOCX)

## Author Contributions

**Conceptualization:** Nelsensius Klau Fauk, Lillian Mwanri, Karen Hawke, Paul Russell Ward.

**Formal analysis:** Nelsensius Klau Fauk.

**Investigation:** Nelsensius Klau Fauk.

**Methodology:** Nelsensius Klau Fauk, Lillian Mwanri, Karen Hawke, Paul Russell Ward.

**Project administration:** Nelsensius Klau Fauk.

**Software:** Nelsensius Klau Fauk.

**Supervision:** Lillian Mwanri, Karen Hawke, Paul Russell Ward.

**Writing – original draft:** Nelsensius Klau Fauk.

**Writing – review & editing:** Nelsensius Klau Fauk, Lillian Mwanri, Karen Hawke, Paul Russell Ward.

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
