## [Decision Letter · Decision Letter 0]

21 Apr 2022

PONE-D-22-03726Traditional treatment of HIV and the role of family members as barriers to access to HIV care service or antiretroviral therapy among people living with HIVPLOS ONE

Dear Dr. Fauk,

Thank you for submitting your manuscript to PLOS ONE. After careful consideration, we feel that it has merit but does not fully meet PLOS ONE’s publication criteria as it currently stands. Therefore, we invite you to submit a revised version of the manuscript that addresses the points raised during the review process.

We look forward to receiving your revised manuscript.

Kind regards,

Bronwyn Myers

Academic Editor

PLOS ONE

Journal Requirements:

Reviewers' comments:

Reviewer's Responses to Questions

**Comments to the Author**

1. Is the manuscript technically sound, and do the data support the conclusions?

Reviewer #1: Yes

Reviewer #2: Yes

2. Has the statistical analysis been performed appropriately and rigorously? 

Reviewer #1: N/A

Reviewer #2: N/A

3. Have the authors made all data underlying the findings in their manuscript fully available?

Reviewer #1: No

Reviewer #2: No

4. Is the manuscript presented in an intelligible fashion and written in standard English?

Reviewer #1: No

Reviewer #2: No

5. Review Comments to the Author

Reviewer #1: Traditional treatment of HIV and the role of family members as barriers to access to HIV care service or antiretroviral therapy among people living with HIV

Use of abbreviations like HIV in the title is not recommended and the title is not well articulated.

- Possibly “Traditional treatment of Human Immunodeficiency Virus as barriers to access HIV care and treatment in Belu, Indonesia”.

- Still can well articulate the title…

General: The paper raised an important issue that is the use of traditional treatments in Belu is a barrier to HIV care and ART. The role players for decision-makers for use of traditional medicine (TM) were family members and the culture. The impact of this practice is to the extent that the patients loss their life. However, this important finding is not well written and elaborated to show how much the use of traditional medicine in these specific areas is affecting the life of HIV patients. The paper is poorly written, I recommend this better be rewritten clearly organizing the drivers for the use TM and its impacts on the life of patients. In writing the manuscript please avoid unnecessary long statements (some of which seems desire to extend the statement – better use short sentences)

- As a researcher you seem biased, because you started by having negative belief on traditional medicine use. It would have been better if the impact on traditional medicine use on access to biomedical HIV care & treatment was assessed.

- Lacks the list of abbreviations/acronyms

- Good to have operational definitions for some terminologies. (like TM, traditional healers … in your context)

Abstract:

- “As a part of a qualitative study in Belu, this paper describes the use of traditional treatment and the role of families in determining traditional treatment for their HIV-positive family member as barriers to access to HIV care service or ART among PLHIV.” This statement not clear better to rewrite in more clear way.

- Snow ball technique you used is not clear – what is the information pack?

Background

- Good coverage, however most sentences are too long that need to be rephrased.

Last paragraph: In this aim statement and throughout the manuscript you have used terms in different words like “HIV care services or ART”, “traditional treatment or medicines” which you could use either of them or better expressive phrase.

- Why you did not discuss on the barriers to access ART, rather than having too narrow scope that focus only on family impact.

Methods:

Study Setting:

Recruitment and Data Collection:

• P1. “As part of a larger qualitative study to understand HIV risk factors, impacts and determinants of access to HIV care services among women and men living with HIV in Belu, this paper describes the use of traditional treatment for HIV and the role of family members, friends and neighbours in supporting the use of traditional treatment as barriers to the access of PLHIV to HIV care service or ART”.

- The aim of this study was already described in the background section; not necessary to repeat it here (remove this statement).

• The snowball technique used is not clear. I do not think this is a snowball technique?

• What was the study information pack given to the patients and HCPs?

• … “HIV clinic receptionist and healthcare facilities” not clear.

P1L11: “The initially interviewed participants were also asked to distribute the study information packs to their friends and colleagues who might be willing to participate.”

- I think this information sharing between participants makes the second group of participants affected (influenced) by others resulting in biased information.

• The method section should have clearly shown the positionality of the researcher (criticality), validity… quality measure of a qualitative study.

Result:

Generally good. Rephrase the statements to sound scientific.

• Table 1: The occupation for HCPs and PLHIV need to be separate. Also, the list of occupation needs to be re-categorized.

Discussion:

• Almost the result is rewritten. It is better to make well-referenced and explained comparing with national or international policies and research findings.

• P1L1: different form referencing (UNAIDS 2020). Make uniform.

Reviewer #2: This was an interesting article. There are some aspects that require revision. First, the entire paper needs to be professionally edited for English language. Many of the sentences are long and the english is hard to follow.

Introduction: I recommend describing barriers to use of ART rather than barriers to accessing HIV care services or use of ART- this is just too wordy and hard to follow.

Please describe the role and use of traditional medicines in Indonesia- more information is needed on the context of these medicines and how traditional and western systems of care for HIV interact if at all.

Methods: The sampling approach you mention here is not snowball sampling- rather convenience sampling

Data analysis section- it very long and can be abbreviated

All the COREQ guidelines have not been followed- please check and report back for example on member checking

Results: You only need 1-2 illustrative quotes per section. Three is too many. There seems to be some overlap between the themes - eg family influence permeates across several themes

Discussion. This needs the most work. You report the findings but do not discuss the implications for policy and practice. If family is so influential what can be done to improve HIV health literacy, overcome medication myths etc, can traditional healers be engaged in the HIV care system to advocate for use of ART? Needs to move beyond mere reporting of the results.

6. PLOS authors have the option to publish the peer review history of their article (what does this mean?). If published, this will include your full peer review and any attached files.

Reviewer #1: No

Reviewer #2: No

---

## [Author Response · Author response to Decision Letter 0]

22 May 2022

Response to reviewer file is attached.

---

## [Decision Letter · Decision Letter 1]

12 Jul 2022

Traditional Human Immunodeficiency Virus treatment and family and social influence as barriers to accessing HIV care services in Belu, Indonesia

PONE-D-22-03726R1

Dear Dr. Fauk,

We’re pleased to inform you that your manuscript has been judged scientifically suitable for publication and will be formally accepted for publication once it meets all outstanding technical requirements.

Kind regards,

Bronwyn Myers

Academic Editor

PLOS ONE

Additional Editor Comments (optional):

Reviewers' comments:

Reviewer's Responses to Questions

**Comments to the Author**

1. If the authors have adequately addressed your comments raised in a previous round of review and you feel that this manuscript is now acceptable for publication, you may indicate that here to bypass the “Comments to the Author” section, enter your conflict of interest statement in the “Confidential to Editor” section, and submit your "Accept" recommendation.

Reviewer #2: All comments have been addressed

2. Is the manuscript technically sound, and do the data support the conclusions?

Reviewer #2: (No Response)

3. Has the statistical analysis been performed appropriately and rigorously? 

Reviewer #2: (No Response)

4. Have the authors made all data underlying the findings in their manuscript fully available?

Reviewer #2: (No Response)

5. Is the manuscript presented in an intelligible fashion and written in standard English?

Reviewer #2: (No Response)

6. Review Comments to the Author

Reviewer #2: (No Response)

7. PLOS authors have the option to publish the peer review history of their article (what does this mean?). If published, this will include your full peer review and any attached files.

Reviewer #2: No

---

## [Editor Report · Acceptance letter]

15 Jul 2022

PONE-D-22-03726R1 

Traditional Human Immunodeficiency Virus treatment and family and social influence as barriers to accessing HIV care services in Belu, Indonesia 

Dear Dr. Fauk:

I'm pleased to inform you that your manuscript has been deemed suitable for publication in PLOS ONE. Congratulations! Your manuscript is now with our production department. 

Kind regards, 

on behalf of

Dr. Bronwyn Myers 

Academic Editor

PLOS ONE